# Structured Bayesian Pruning via Log-Normal Multiplicative Noise

**Kirill Neklyudov** [1,2]     **Dmitry Molchanov** [1,3]     **Arsenii Ashukha** [1,2]     **Dmitry Vetrov** [1,2]
k.necludov@gmail.com      dmolchanov@hse.ru        aashukha@hse.ru         dvetrov@hse.ru

[1]National Research University Higher School of Economics   [2]Yandex
[3]Skolkovo Institute of Science and Technology

## Abstract

Dropout-based regularization methods can be regarded as injecting random noise with pre-defined magnitude to different parts of the neural network during training. It was recently shown that Bayesian dropout procedure not only improves generalization but also leads to extremely sparse neural architectures by automatically setting the individual noise magnitude per weight. However, this sparsity can hardly be used for acceleration since it is unstructured. In the paper, we propose a new Bayesian model that takes into account the computational structure of neural networks and provides structured sparsity, e.g. removes neurons and/or convolutional channels in CNNs. To do this we inject noise to the neurons outputs while keeping the weights unregularized. We establish the probabilistic model with a proper truncated log-uniform prior over the noise and truncated log-normal variational approximation that ensures that the KL-term in the evidence lower bound is computed in closed-form. The model leads to structured sparsity by removing elements with a low SNR from the computation graph and provides significant acceleration on a number of deep neural architectures. The model is easy to implement as it can be formulated as a separate dropout-like layer.

## 1   Introduction

Deep neural networks are a flexible family of models which provides state-of-the-art results in many machine learning problems [14, 20]. However, this flexibility often results in overfitting. A common solution for this problem is regularization. One of the most popular ways of regularization is Binary Dropout [19] that prevents co-adaptation of neurons by randomly dropping them during training. An equally effective alternative is Gaussian Dropout [19] that multiplies the outputs of the neurons by Gaussian random noise. In recent years several Bayesian generalizations of these techniques have been developed, e.g. Variational Dropout [8] and Variational Spike-and-Slab Neural Networks [13]. These techniques provide theoretical justification of different kinds of Dropout and also allow for automatic tuning of dropout rates, which is an important practical result.

Besides overfitting, compression and acceleration of neural networks are other important challenges, especially when memory or computational resources are restricted. Further studies of Variational Dropout show that individual dropout rates for each weight allow to shrink the original network architecture and result in a highly sparse model [16]. General sparsity provides a way of neural network compression, while the time of network evaluation may remain the same, as most modern DNN-oriented software can't work with sparse matrices efficiently. At the same time, it is possible to achieve acceleration by enforcing *structured* sparsity in convolutional filters or data tensors. In the simplest case it means removing redundant neurons or convolutional filters instead of separate weights; but more complex patterns can also be considered. This way Group-wise Brain Damage

[10] employs group-wise sparsity in convolutional filters, Perforated CNNs [3] drop redundant rows from the intermediate dataframe matrices that are used to compute convolutions, and Structured Sparsity Learning [24] provides a way to remove entire convolutional filters or even layers in residual networks. These methods allow to obtain practical acceleration with little to no modifications of the existing software. In this paper, we propose a tool that is able to induce an arbitrary pattern of structured sparsity on neural network parameters or intermediate data tensors. We propose a dropout-like layer with a parametric multiplicative noise and use stochastic variational inference to tune its parameters in a Bayesian way. We introduce a proper analog of sparsity-inducing log-uniform prior distribution [8, 16] that allows us to formulate a correct probabilistic model and avoid the problems that come from using an improper prior. This way we obtain a novel Bayesian method of regularization of neural networks that results in structured sparsity. Our model can be represented as a separate dropout-like layer that allows for a simple and flexible implementation with almost no computational overhead, and can be incorporated into existing neural networks.

Our experiments show that our model leads to high group sparsity level and significant acceleration of convolutional neural networks with negligible accuracy drop. We demonstrate the performance of our method on LeNet and VGG-like architectures using MNIST and CIFAR-10 datasets.

## 2   Related Work

Deep neural networks are extremely prone to overfitting, and extensive regularization is crucial. The most popular regularization methods are based on injection of multiplicative noise over layer inputs, parameters or activations [8, 19, 22]. Different kinds of multiplicative noise have been used in practice; the most popular choices are Bernoulli and Gaussian distributions. Another type of regularization of deep neural networks is based on reducing the number of parameters. One approach is to use low-rank approximations, e.g. tensor decompositions [4, 17], and the other approach is to induce sparsity, e.g. by pruning [5] or $L_1$ regularization [24]. Sparsity can also be induced by using the Sparse Bayesian Learning framework with empirical Bayes [21] or with sparsity-inducing priors [12, 15, 16].

High sparsity is one of the key factors for the compression of DNNs [5, 21]. However, in addition to compression it is beneficial to obtain acceleration. Recent papers propose different approaches to acceleration of DNNs, e.g. Spatial Skipped Convolutions [3] and Spatially Adaptive Computation Time [2] that propose different ways to reduce the number of computed convolutions, Binary Networks [18] that achieve speedup by using only 1 bit to store a single weight of a DNN, Low-Rank Expansions [6] that use low-rank filter approximations, and Structured Sparsity Learning [24] that allows to remove separate neurons or filters. As reported in [24] it is possible to obtain acceleration of DNNs by introducing structured sparsity, e.g. by removing whole neurons, filters or layers. However, non-adaptive regularization techniques require tuning of a huge number of hyperparameters that makes it difficult to apply in practice. In this paper we apply the Bayesian learning framework to obtain structured sparsity and focus on acceleration of neural networks.

## 3   Stochastic Variational Inference

Given a probabilistic model $p(y \,|\, x, \theta)$ we want to tune parameters $\theta$ of the model using training dataset $\mathcal{D} = \{(x_i, y_i)\}_{i=1}^N$. The prior knowledge about parameters $\theta$ is defined by prior distribution $p(\theta)$. Using the Bayes rule we obtain the posterior distribution $p(\theta \,|\, \mathcal{D}) = p(\mathcal{D} \,|\, \theta)p(\theta)/p(\mathcal{D})$. However, computing posterior distribution using the Bayes rule usually involves computation of intractable integrals, so we need to use approximation techniques.

One of the most widely used approximation techniques is *Variational Inference*. In this approach the unknown distribution $p(\theta \,|\, \mathcal{D})$ is approximated by a parametric distribution $q_\phi(\theta)$ by minimization of the Kullback-Leibler divergence $\mathrm{KL}(q_\phi(\theta) \,\|\, p(\theta \,|\, \mathcal{D}))$. Minimization of the KL divergence is equivalent to maximization of the *variational lower bound* $\mathcal{L}(\phi)$.

$$\mathcal{L}(\phi) = L_\mathcal{D}(\phi) - \mathrm{KL}(q_\phi(\theta) \,\|\, p(\theta)), \qquad (1)$$

$$\text{where } L_\mathcal{D}(\phi) = \sum_{i=1}^N \mathbb{E}_{q_\phi(\theta)} \log p(y_i \,|\, x_i, \theta) \qquad (2)$$

$L_D(\phi)$ is a so-called expected log-likelihood function which is intractable in case of complex probabilistic model $p(y \mid x, \theta)$. Following [8] we use the Reparametrization trick to obtain an unbiased differentiable minibatch-based Monte Carlo estimator of the expected log-likelihood. Here $N$ is the total number of objects, $M$ is the minibatch size, and $f(\phi, \varepsilon)$ provides samples from the approximate posterior $q_\phi(\theta)$ as a deterministic function of a non-parametric noise $\varepsilon \sim p(\varepsilon)$.

$$L_D(\phi) \simeq L_D^{SGVB}(\phi) = \frac{N}{M} \sum_{k=1}^{M} \log p(y_{i_k} \mid x_{i_k}, w_{i_k} = f(\phi, \varepsilon_{i_k})) \tag{3}$$

$$\mathcal{L}(\phi) \simeq \mathcal{L}^{SGVB}(\phi) = L_D^{SGVB}(\phi) - \mathrm{KL}(q_\phi(w) \,\|\, p(w)) \tag{4}$$

$$\nabla_\phi L_D(\phi) \simeq \nabla_\phi L_D^{SGVB}(\phi) \tag{5}$$

This way we obtain a procedure of approximate Bayesian inference where we solve optimization problem (4) by stochastic gradient ascent w.r.t. variational parameters $\phi$. This procedure can be efficiently applied to Deep Neural Networks and usually the computational overhead is very small, as compared to ordinary DNNs.

If the model $p(y \mid x, \theta, w)$ has another set of parameters $w$ that we do not want to be Bayesian about, we can still use the same variational lower bound objective:

$$\mathcal{L}(\phi, w) = L_D(\phi, w) - \mathrm{KL}(q_\phi(\theta) \,\|\, p(\theta)) \to \max_{\phi, w}, \tag{6}$$

$$\text{where } L_D(\phi, w) = \sum_{i=1}^{N} \mathbb{E}_{q_\phi(\theta)} \log p(y_i \mid x_i, \theta, w) \tag{7}$$

This objective corresponds the maximum likelihood estimation $w_{ML}$ of parameters $w$, while finding the approximate posterior distribution $q_\phi(\theta) \approx p(\theta \mid \mathcal{D}, w_{ML})$. In this paper we denote the weights of the neural networks, the biases, etc. as $w$ and find their maximum likelihood estimation as described above. The parameters $\theta$ that undergo the Bayesian treatment are the noisy masks in the proposed dropout-like layer (SBP layer). They are described in the following section.

## 4    Group Sparsity with Log-normal Multiplicative Noise

Variational Inference with a sparsity-inducing log-uniform prior over the weights of a neural network is an efficient way to enforce general sparsity on weight matrices [16]. However, it is difficult to apply this approach to explicitly enforce structured sparsity. We introduce a dropout-like layer with a certain kind of multiplicative noise. We also make use of the sparsity-inducing log-uniform prior, but put it over the noise variables rather than weights. By sharing those noise variables we can enforce group-wise sparsity with any form of groups.

### 4.1    Variational Inference for Group Sparsity Model

We consider a single dropout-like layer with an input vector $x \in \mathbb{R}^I$ that represents one object with $I$ features, and an output vector $y \in \mathbb{R}^I$ of the same size. The input vector $x$ is usually supposed to come from the activations of the preceding layer. The output vector $y$ would then serve as an input vector for the following layer. We follow the general way to build dropout-like layers (8). Each input feature $x_i$ is multiplied by a noise variable $\theta_i$ that comes from some distribution $p_{noise}(\theta)$. For example, for Binary Dropout $p_{noise}(\theta)$ would be a fully factorized Bernoulli distribution with $p_{noise}(\theta_i) = \mathrm{Bernoulli(p)}$, and for Gaussian dropout it would be a fully-factorized Gaussian distribution with $p_{noise}(\theta_i) = \mathcal{N}(1, \alpha)$.

$$y_i = x_i \cdot \theta_i \qquad\qquad \theta \sim p_{noise}(\theta) \tag{8}$$

Note that if we have a minibatch $X^{M \times I}$ of $M$ objects, we would independently sample a separate noise vector $\theta^m$ for each object $x^m$. This would be the case throughout the paper, but for the sake of simplicity we would consider a single object $x$ in all following formulas. Also note that the noise $\theta$ is usually only sampled during the training phase. A common approximation during the testing phase is to use the expected value $\mathbb{E}\theta$ instead of sampling $\theta$. All implementation details are provided and discussed in Section 4.5.

We follow a Bayesian treatment of the variable $\theta$, as described in Section 3. In order to obtain a sparse solution, we choose the prior distribution $p(\theta)$ to be a fully-factorized improper log-uniform distribution. We denote this distribution as $\mathrm{LogU}_\infty(\cdot)$ to stress that it has infinite domain. This distribution is known for its sparsification properties and works well in practice for deep neural networks [16].

$$p(\theta) = \prod_{i=1}^{I} p(\theta_i) \qquad\qquad p(\theta_i) = \mathrm{LogU}_\infty(\theta_i) \propto \frac{1}{\theta_i} \qquad\qquad \theta_i > 0 \qquad (9)$$

In order to train the model, i.e. perform variational inference, we need to choose an approximation family $q_\phi$ for the posterior distribution $p(\theta \,|\, \mathcal{D}) \approx q_\phi(\theta)$.

$$q_\phi(\theta) = \prod_{i=1}^{I} q(\theta_i \,|\, \mu_i, \sigma_i) = \prod_{i=1}^{I} \mathrm{LogN}(\theta_i \,|\, \mu_i, \sigma_i^2) \qquad\qquad (10)$$

$$\theta_i \sim \mathrm{LogN}(\theta_i \,|\, \mu_i, \sigma_i^2) \quad\Longleftrightarrow\quad \log\theta_i \sim \mathcal{N}(\log\theta_i \,|\, \mu_i, \sigma_i^2) \qquad\qquad (11)$$

A common choice of variational distribution $q(\cdot)$ is a fully-factorized Gaussian distribution. However, for this particular model we choose $q(\theta)$ to be a fully-factorized log-normal distribution (10–11). To make this choice, we were guided by the following reasons:

- The log-uniform distribution is a specific case of the log-normal distribution when the parameter $\sigma$ goes to infinity and $\mu$ remains fixed. Thus we can guarantee that in the case of no data our variational approximation can be made exact. Hence this variational family has no "prior gap".

- We consider a model with multiplicative noise. The scale of this noise corresponds to its shift in the logarithmic space. By establishing the log-uniform prior we set no preferences on different scales of this multiplicative noise. The usual use of a Gaussian as a posterior immediately implies very asymmetric skewed distribution in the logarithmic space. Moreover log-uniform and Gaussian distributions have different supports and that will require establishing two log-uniform distributions for positive and negative noises. In this case Gaussian variational approximation would have quite exotic bi-modal form (one mode in the log-space of positive noises and another one in the log-space of negative noises). On the other hand, the log-normal posterior for the multiplicative noise corresponds to a Gaussian posterior for the additive noise in the logarithmic scale, which is much easier to interpret.

- Log-normal noise is always non-negative both during training and testing phase, therefore it does not change the sign of its input. This is in contrast to Gaussian multiplicative noise $\mathcal{N}(\theta_i \,|\, 1, \alpha)$ that is a standard choice for Gaussian dropout and its modifications [8, 19, 23]. During the training phase Gaussian noise can take negative values, so the input to the following layer can be of arbitrary sign. However, during the testing phase noise $\theta$ is equal to 1, so the input to the following layer is non-negative with many popular non-linearities (e.g. ReLU, sigmoid, softplus). Although Gaussian dropout works well in practice, it is difficult to justify notoriously different input distributions during training and testing phases.

- The log-normal approximate posterior is tractable. Specifically, the KL divergence term $\mathrm{KL}(\mathrm{LogN}(\theta \,|\, \mu, \sigma^2) \,\|\, \mathrm{LogU}_\infty(\theta))$ can be computed analytically.

The final loss function is presented in equation (12) and is essentially the original variational lower bound (4).

$$\mathcal{L}^{SGVB}(\phi) = L_D^{SGVB}(\mu, \sigma, W) - \mathrm{KL}(q(\theta \,|\, \mu, \sigma) \,\|\, p(\theta)) \to \max_{\mu, \sigma, W}, \qquad\qquad (12)$$

where $\mu$ and $\sigma$ are the variatianal parameters, and $W$ denotes all other trainable parameters of the neural network, e.g. the weight matrices, the biases, batch normalization parameters, etc.

Note that we can optimize the variational lower bound w.r.t. the parameters $\mu$ and $\sigma$ of the log-normal noise $\theta$. We do not fix the mean of the noise thus making our variational approximation more tight.

## 4.2  Problems of Variational Inference with Improper Log-Uniform Prior

The log-normal posterior in combination with a log-uniform prior has a number of attractive features. However, the maximization of the variational lower bound with a log-uniform prior and a log-normal

posterior is an ill-posed optimization problem. As the log-uniform distribution is an improper prior, the KL-divergence between a log-normal distribution $\text{LogN}(\mu, \sigma^2)$ and a log-uniform distribution $\text{LogU}_\infty$ is infinite for any finite value of parameters $\mu$ and $\sigma$.

$$\text{KL}\left(\text{LogN}(x \,|\, \mu, \sigma^2) \,\|\, \text{LogU}_\infty(x)\right) = C - \log \sigma \qquad\qquad C = +\infty \qquad (13)$$

A common way to tackle this problem is to consider the density of the log-uniform distribution to be equal to $\frac{C}{\theta}$ and to treat $C$ as some finite constant. This trick works well for the case of a Gaussian posterior distribution [8, 16]. The KL divergence between a Gaussian posterior and a log-uniform prior has an infinite gap, but can be calculated up to this infinite constant in a meaningful way [16]. However, for the case of the log-normal posterior the KL divergence is infinite for any finite values of variational parameters, and is equal to zero for a fixed finite $\mu$ and infinite $\sigma$. As the data-term (3) is bounded for any value of variational parameters, the only global optimum of the variational lower bound is achieved when $\mu$ is finite and fixed, and $\sigma$ goes to infinity. In this case the posterior distribution collapses into the prior distribution and the model fails to extract any information about the data. This effect is wholly caused by the fact that the log-uniform prior is an improper (non-normalizable) distribution, which makes the whole probabilistic model flawed.

## 4.3 Variational Inference with Truncated Approximation Family

Due to the improper prior the optimization problem becomes ill-posed. But do we really need to use an improper prior distribution? The most common number format that is used to represent the parameters of a neural network is the *floating-point* format. The *floating-point* format is only able to represent numbers from a limited range. For example, a single-point precision variable can only represent numbers from the range $-3.4 \times 10^{38}$ to $+3.4 \times 10^{38}$, and the smallest possible positive number is equal to $1.2 \times 10^{-38}$. All of probability mass of the improper log-uniform prior is concentrated beyond the single-point precision (and essentially any practical floating point precision), not to mention that the actual relevant range of values of neural network parameters is much smaller. It means that in practice this prior is not a good choice for software implementation of neural networks.

We propose to use a truncated log-uniform distribution (14) as a proper analog of the log-uniform distribution. Here $I_{[a,b]}(x)$ denotes the indicator function for the interval $x \in [a, b]$. The posterior distribution should be defined on the same support as the prior distribution, so we also need to use a *truncated log-normal* distribution (14).

$$\text{LogU}_{[a,b]}(\theta_i) \propto \text{LogU}_\infty(\theta_i) \cdot I_{[a,b]}(\log \theta_i) \qquad \text{LogN}_{[a,b]}(\theta_i) \propto \text{LogN}(\theta_i \,|\, \mu_i, \sigma_i^2) \cdot I_{[a,b]}(\log \theta_i)$$
$$(14)$$

Our final model then can be formulated as follows.

$$y_i = x_i \cdot \theta_i \qquad p(\theta_i) = \text{LogU}_{[a,b]}(\theta_i) \qquad q(\theta_i \,|\, \mu_i, \sigma_i) = \text{LogN}_{[a,b]}(\theta_i \,|\, \mu_i, \sigma_i^2) \qquad (15)$$

Note that all the nice facts about the log-normal posterior distribution from the Section 4.1 are also true for the truncated log-normal posterior. However, now we have a proper probabilistic model and the Stochastic Variational Inference can be preformed correctly. Unlike (13), now the KL divergence term (16–17) can be calculated correctly for all valid values of variational parameters (see Appendix A for details).

$$\text{KL}(q(\theta \,|\, \mu, \sigma) \,\|\, p(\theta)) = \sum_{i=1}^{I} \text{KL}(q(\theta_i \,|\, \mu_i, \sigma_i) \,\|\, p(\theta_i)) \qquad (16)$$

$$\text{KL}(q(\theta_i \,|\, \mu_i, \sigma_i) \,\|\, p(\theta_i)) = \log \frac{b - a}{\sqrt{2\pi e \sigma_i^2}} - \log(\Phi(\beta_i) - \Phi(\alpha_i)) - \frac{\alpha_i \phi(\alpha_i) - \beta_i \phi(\beta_i)}{2(\Phi(\beta_i) - \Phi(\alpha_i))}, \qquad (17)$$

where $\alpha_i = \frac{a - \mu_i}{\sigma_i}$, $\beta_i = \frac{b - \mu_i}{\sigma_i}$, $\phi(\cdot)$ and $\Phi(\cdot)$ are the density and the CDF of the standard normal distribution.

The reparameterization trick also can still be performed (18) using the inverse CDF of the truncated normal distribution (see Appendix B).

$$\theta_i = \exp\left(\mu_i + \sigma_i \Phi^{-1}\left(\Phi(\alpha_i) + (\Phi(\beta_i) - \Phi(\alpha_i)) y_i\right)\right), \text{ where } y_i \sim \mathcal{U}(y \,|\, 0, 1) \qquad (18)$$

The final loss and the set of parameters is the same as described in Section 4.1, and the training procedure remains the same.

## 4.4 Sparsity

Log-uniform prior is known to lead to a sparse solution [16]. In the variational dropout paper authors interpret the parameter $\alpha$ of the multiplicative noise $\mathcal{N}(1, \alpha)$ as a Gaussian dropout rate and use it as a thresholding criterion for weight pruning. Unlike the binary or Gaussian dropout, in the truncated log-normal model there is no "dropout rate" variable. However, we can use the signal-to-noise ratio $\mathbb{E}\theta/\sqrt{\mathrm{Var}(\theta)}$ (SNR) for thresholding.

$$\mathrm{SNR}(\theta_i) = \frac{(\Phi(\sigma_i - \alpha_i) - \Phi(\sigma_i - \beta_i))/\sqrt{\Phi(\beta_i) - \Phi(\alpha_i)}}{\sqrt{\exp(\sigma_i^2)(\Phi(2\sigma_i - \alpha_i) - \Phi(2\sigma_i - \beta_i)) - (\Phi(\sigma_i - \alpha_i) - \Phi(\sigma_i - \beta_i))^2}} \tag{19}$$

The SNR can be computed analytically, the derivation can be found in the appendix. It has a simple interpretation. If the SNR is low, the corresponding neuron becomes very noisy and its output no longer contains any useful information. If the SNR is high, it means that the neuron output contains little noise and is important for prediction. Therefore we can remove all neurons or filters with a low SNR and set their output to constant zero.

## 4.5 Implementation details

We perform a minibatch-based stochastic variational inference for training. The training procedure looks as follows. On each training step we take a minibatch of $M$ objects and feed it into the neural network. Consider a single SBP layer with input $X^{M \times I}$ and output $Y^{M \times I}$. We independently sample a separate noise vector $\theta^m \sim q(\theta)$ for each object $x_m$ and obtain a noise matrix $\theta^{M \times I}$. The output matrix $Y^{M \times I}$ is then obtained by component-wise multiplication of the input matrix and the noise matrix: $y_{mi} = x_{mi} \cdot \theta_i^m$.

To be fully Bayesian, one would also sample and average over different dropout masks $\theta$ during testing, i.e. perform Bayesian ensembling. Although this procedure can be used to slightly improve the final accuracy, it is usually avoided. Bayesian ensembling essentially requires sampling of different copies of neural networks, which makes the evaluation $K$ times slower for averaging over $K$ samples. Instead, during the testing phase in most dropout-based techniques the noise variable $\theta$ is replaced with its expected value. In this paper we follow the same approach and replace all non-pruned $\theta_i$ with their expectations (20) during testing. The derivation of the expectation of the truncated log-normal distribution is presented in Appendix C.

$$\mathbb{E}\theta_i = \frac{\exp(\mu_i + \sigma_i^2/2)}{\Phi(\beta_i) - \Phi(\alpha_i)} \left[ \Phi\left(\frac{\sigma_i^2 + \mu_i - a}{\sigma_i}\right) - \Phi\left(\frac{\sigma_i^2 + \mu_i - b}{\sigma_i}\right) \right] \tag{20}$$

We tried to use Bayesian ensembling with this model, and experienced almost no gain of accuracy. It means that the variance of the learned approximate posterior distribution is low and does not provide a rich ensemble.

Throughout the paper we introduced the SBP dropout layer for the case when input objects are represented as one-dimensional vectors $x$. When defined like that, it would induce general sparsity on the input vector $x$. It works as intended for fully-connected layers, as a single input feature corresponds to a single output neuron of a preceding fully-connected layer and a single output neuron of the following layer. However, it is possible to apply the SBP layer in a more generic setting. Firstly, if the input object is represented as a multidimensional tensor $X$ with shape $I_1 \times I_2 \times \cdots \times I_d$, the noise vector $\theta$ of length $I = I_1 \times I_2 \times \cdots \times I_d$ can be reshaped into a tensor with the same shape. Then the output tensor $Y$ can be obtained as a component-wise product of the input tensor $X$ and the noise tensor $\theta$. Secondly, the SBP layer can induce any form of structured sparsity on this input tensor $X$. To do it, one would simply need to use a single random variable $\theta_i$ for the group of input features that should be removed simultaneously. For example, consider an input tensor $X^{H \times W \times C}$ that comes from a convolutional layer, $H$ and $W$ being the size of the image, and $C$ being the number of channels. Then, in order to remove redundant filters from the preceding layer (and at the same time redundant channels from the following layer), one need to share the random variables $\theta$ in the following way:

$$y_{hwc} = x_{hwc} \cdot \theta_c \qquad\qquad \theta_c \sim \mathrm{LogN}_{[a,b]}(\theta_c \,|\, \mu_c, \sigma_c^2) \tag{21}$$

Note that now there is one sample $\theta \in \mathbb{R}^C$ for one object $X^{H \times W \times C}$ on each training step. If the signal-to-noise ratio becomes lower than 1 for a component $\theta_c$, that would mean that we can

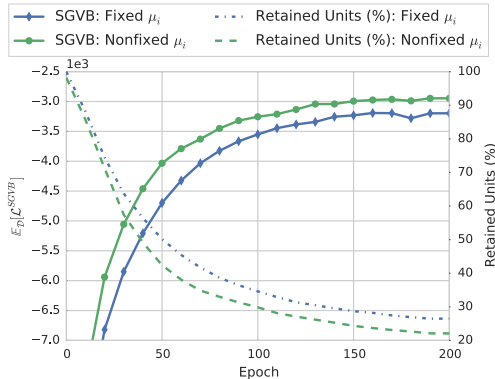

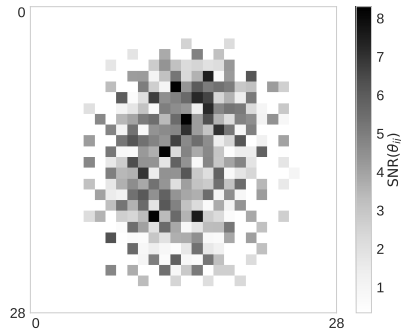

Figure 1: The value of the SGVB for the case of fixed variational parameter $\mu = 0$ (blue line) and for the case when both variational parameters $\mu$ and $\sigma$ are trained (green line)

Figure 2: The learned signal-to-noise ratio for image features on the MNIST dataset.

permanently remove the $c$-th channel of the input tensor, and therefore delete the $c$-th filter from the preceding layer and the $c$-th channel from the following layer. All the experiments with convolutional architectures used this formulation of SBP. This is a general approach that is not limited to reducing the shape of the input tensor. It is possible to obtain any fixed pattern of group-wise sparsity using this technique.

Similarly, the SBP layer can be applied in a DropConnect fashion. One would just need to multiply the weight tensor $W$ by a noise tensor $\theta$ of similar shape. The training procedure remains the same. It is still possible to enforce any structured sparsity pattern for the weight tensor $W$ by sharing the random variables as described above.

## 5 Experiments

We perform an evaluation on different supervised classification tasks and with different architectures of neural networks including deep VGG-like architectures with batch normalization layers. For each architecture, we report the number of retained neurons and filters, and obtained acceleration. Our experiments show that *Structured Bayesian Pruning* leads to a high level of structured sparsity in convolutional filters and neurons of DNNs without significant accuracy drop. We also demonstrate that optimization w.r.t. the full set of variational parameters $(\mu, \sigma)$ leads to improving model quality and allows us to perform sparsification in a more efficient way, as compared to tuning of only one free parameter that corresponds to the noise variance. As a nice bonus, we show that *Structured Bayesian Pruning* network does not overfit on randomly labeled data, that is a common weakness of non-bayesian dropout networks. The source code is available in Theano [7] and Lasagne, and also in TensorFlow [1] (`https://github.com/necludov/group-sparsity-sbp`).

### 5.1 Experiment Setup

The truncation parameters $a$ and $b$ are the hyperparameters of our model. As our layer is meant for regularization of the model, we would like our layer not to amplify the input signal and restrict the noise $\theta$ to an interval $[0, 1]$. This choice corresponds to the right truncation threshold $b$ set to $0$. We find empirically that the left truncation parameter $a$ does not influence the final result much. We use values $a = -20$ and $b = 0$ in all experiments.

We define redundant neurons by the signal-to-noise ratio of the corresponding multiplicative noise $\theta$. See Section 4.4 for more details. By removing all neurons and filters with the $\text{SNR} < 1$ we experience no accuracy drop in all our experiments. SBP dropout layers were put *after* each convolutional layer to remove its filters, and *before* each fully-connected layer to remove its input neurons. As one filter of the last convolutional layer usually corresponds to a group of neurons in the following dense layer, it means that we can remove more input neurons in the first dense layer. Note that it means that we have two consecutive dropout layers between the last convolutional layer and the first fully-connected layer in CNNs, and a dropout layer before the first fully-connected layer in FC networks (see Fig. 2).

Table 1: Comparison of different structured sparsity inducing techniques on LeNet-5-Caffe and LeNet-500-300 architectures. SSL [24] is based on group lasso regularization, SparseVD [16]) is a Bayesian model with a log-uniform prior that induces weight-wise sparsity. For SparseVD a neuron/filter is considered pruned, if all its weights are set to 0. Our method provides the highest speed-up with a similar accuracy. We report acceleration that was computed on CPU (Intel Xeon E5-2630), GPU (Tesla K40) and in terms of Floating Point Operations (FLOPs).

| Network | Method | Error % | Neurons per Layer | CPU | GPU | FLOPs |
|---|---|---|---|---|---|---|
| | Original | 1.54 | $784 - 500 - 300 - 10$ | $1.00\times$ | $1.00\times$ | $1.00\times$ |
| | SparseVD | 1.57 | $537 - 217 - 130 - 10$ | $1.19\times$ | $1.03\times$ | $3.73\times$ |
| LeNet-500-300 | SSL | 1.49 | $434 - 174 - \;\;78 - 10$ | $2.21\times$ | $1.04\times$ | $6.06\times$ |
| (ours) | StructuredBP | 1.55 | $245 - 160 - \;\;55 - 10$ | $2.33\times$ | $1.08\times$ | $11.23\times$ |
| | Original | 0.80 | $20 - 50 - 800 - 500$ | $1.00\times$ | $1.00\times$ | $1.00\times$ |
| | SparseVD | 0.75 | $17 - 32 - 329 - \;\;75$ | $1.48\times$ | $1.41\times$ | $2.19\times$ |
| LeNet5-Caffe | SSL | 1.00 | $3 - 12 - 800 - 500$ | $5.17\times$ | $1.80\times$ | $3.90\times$ |
| (ours) | StructuredBP | 0.86 | $3 - 18 - 284 - 283$ | $5.41\times$ | $1.91\times$ | $10.49\times$ |

Table 2: Comparison of different structured sparsity inducing techniques (SparseVD [16]) on VGG-like architectures on CIFAR-10 dataset. StructuredBP stands for the original SBP model, and StructuredBPa stands for the SBP model with KL scaling. $k$ is a width scale factor that determines the number of neurons or filters on each layer of the network (width$(k) = k \times$ original width)

| k | Method | Error % | Units per Layer | CPU | GPU | FLOPs |
|---|---|---|---|---|---|---|
| 1.0 | Original | 7.2 | $64 - 64 - 128 - 128 - 256 - 256 - 256 - 512 - 512 - 512 - 512 - 512 - 512 - 512$ | $1.00\times$ | $1.00\times$ | $1.00\times$ |
| | SparseVD | 7.2 | $64 - 62 - 128 - 126 - 234 - 155 - \;\;31 - \;\;81 - \;\;76 - \;\;\;\;9 - 138 - 101 - 413 - 373$ | $2.50\times$ | $1.69\times$ | $2.27\times$ |
| (ours) | StructuredBP | 7.5 | $64 - 62 - 128 - 126 - 234 - 155 - \;\;31 - \;\;79 - \;\;73 - \;\;\;\;9 - \;\;59 - \;\;73 - \;\;56 - \;\;27$ | $2.71\times$ | $1.74\times$ | $2.30\times$ |
| (ours) | StructuredBPa | 9.0 | $44 - 54 - \;\;92 - 115 - 234 - 155 - \;\;31 - \;\;76 - \;\;55 - \;\;\;\;9 - \;\;34 - \;\;35 - \;\;21 - 280$ | $3.68\times$ | $2.06\times$ | $3.16\times$ |
| 1.5 | Original | 6.8 | $96 - 96 - 192 - 192 - 384 - 384 - 384 - 768 - 768 - 768 - 768 - 768 - 768 - 768$ | $1.00\times$ | $1.00\times$ | $1.00\times$ |
| | SparseVD | 7.0 | $96 - 78 - 191 - 146 - 254 - 126 - \;\;27 - \;\;79 - \;\;74 - \;\;\;\;9 - 137 - 100 - 416 - 479$ | $3.35\times$ | $2.16\times$ | $3.27\times$ |
| (ours) | StructuredBP | 7.2 | $96 - 77 - 190 - 146 - 254 - 126 - \;\;26 - \;\;79 - \;\;70 - \;\;\;\;9 - \;\;71 - \;\;82 - \;\;79 - \;\;49$ | $3.63\times$ | $2.17\times$ | $3.32\times$ |
| (ours) | StructuredBPa | 7.8 | $77 - 74 - 161 - 146 - 254 - 125 - \;\;26 - \;\;78 - \;\;66 - \;\;\;\;9 - \;\;47 - \;\;55 - \;\;54 - 237$ | $4.47\times$ | $2.47\times$ | $3.93\times$ |

## 5.2 More Flexible Variational Approximation

Usually during automatic training of dropout rates the mean of the noise distribution remains fixed. In the case of our model it is possible to train both mean and variance of the multiplicative noise. By using a more flexible distribution we obtain a tighter variational lower bound and a higher sparsity level. In order to demonstrate this effect, we performed an experiment on MNIST dataset with a fully connected neural network that contains two hidden layers with 1000 neurons each. The results are presented in Fig. 1.

## 5.3 LeNet5 and Fully-Connected Net on MNIST

We compare our method with other sparsity inducing methods on the MNIST dataset using a fully connected architecture LeNet-500-300 and a convolutional architecture LeNet-5-Caffe. These networks were trained with Adam without any data augmentation. The LeNet-500-300 network was trained from scratch, and the LeNet-5-Caffe[1] network was pretrained with weight decay. An illustration of trained SNR for the image features for the LeNet-500-300[2] network is shown in Fig. 2. The final accuracy, group-wise sparsity levels and speedup for these architectures for different methods are shown in Table 1.

## 5.4 VGG-like on CIFAR-10

To prove that SBP scales to deep architectures, we apply it to a VGG-like network [25] that was adapted for the CIFAR-10 [9] dataset. The network consists of 13 convolutional and two fully-connected layers, trained with pre-activation batch normalization and Binary Dropout. At the start of the training procedure, we use pre-trained weights for initialization. Results with different scaling of the number of units are presented in Table 2. We present results for two architectures with different scaling coefficient $k \in \{1.0, 1.5\}$ . For smaller values of scaling coefficient $k \in \{0.25, 0.5\}$ we obtain less sparse architecture since these networks have small learning capacities. Besides the results for the standard StructuredBP procedure, we also provide the results for SBP with KL scaling (StructuredBPa). Scaling the KL term of the variational lower bound proportional to the computational complexity of the layer leads to a higher sparsity level for the first layers, providing

more acceleration. Despite the higher error values, we obtain the higher value of true variational lower bound during KL scaling, hence, we find its another local maximum.

### 5.5 Random Labels

A recent work shows that Deep Neural Networks have so much capacity that they can easily memorize the data even with random labeling [26]. Binary dropout as well as other standard regularization techniques do not prevent the networks from overfitting in this scenario. However, recently it was shown that Bayesian regularization may help [16]. Following these works, we conducted similar experiments. We used a Lenet5 network on the MNIST dataset and a VGG-like network on CIFAR-10. Although Binary Dropout does not prevent these networks from overfitting, SBP decides to remove all neurons of the neural network and provides a constant prediction. In other words, in this case SBP chooses the simplest model that achieves the same testing error rate. This is another confirmation that Bayesian regularization is more powerful than other popular regularization techniques.

## 6 Conclusion

We propose Structured Bayesian Pruning, or SBP, a dropout-like layer that induces multiplicative random noise over the output of the preceding layer. We put a sparsity-inducing prior over the noise variables and tune the noise distribution using stochastic variational inference. SBP layer can induce an arbitrary structured sparsity pattern over its input and provides adaptive regularization. We apply SBP to cut down the number of neurons and filters in convolutional neural networks and report significant practical acceleration with no modification of the existing software implementation of these architectures.

### Acknowledgments

We would like to thank Christos Louizos and Max Welling for valuable discussions. Kirill Neklyudov and Arsenii Ashukha were supported by HSE International lab of Deep Learning and Bayesian Methods which is funded by the Russian Academic Excellence Project '5-100'. Dmitry Molchanov was supported by the Ministry of Education and Science of the Russian Federation (grant 14.756.31.0001). Dmitry Vetrov was supported by the Russian Science Foundation grant 17-11-01027.

## Footnotes

[1]A modified version of LeNet5 from [11]. Caffe Model specification: `https://goo.gl/4yI3dL`

[2]Fully Connected Neural Net with 2 hidden layers that contains 500 and 300 neurons respectively.

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
