[Supplementary Material · appendix.pdf]

## A    KL divergence for the truncated log-normal distribution

We need to introduce some notation to work with the truncated normal distribution. Consider a distribution $\mathrm{t}\mathcal{N}(x \,|\, \mu, \sigma^2, a, b)$, where $\mu$ and $\sigma$ are the mean and the standard deviation of the corresponding normal distribution before truncation, and $a$ and $b$ are the left and right truncation thresholds respectively. Denote

$$\alpha = \frac{a - \mu}{\sigma} \qquad \beta = \frac{b - \mu}{\sigma} \qquad Z = \Phi(\beta) - \Phi(\alpha)$$

Now we can calculate the KL divergence between a truncated log-normal distribution $q(\theta^i)$ and a log-uniform distribution $p(\theta^i)$ with a bounded support $\theta^i \in [e^a, e^b]$.

$$\mathrm{KL}(q(\theta^i) \,\|\, p(\theta^i)) = \mathrm{KL}(q(\log \theta^i) \,\|\, p(\log \theta^i)) =$$

$$= \mathrm{KL}(\mathrm{t}\mathcal{N}(x \,|\, \mu, \sigma^2, a, b) \,\|\, \mathcal{U}(x \,|\, a, b)) = \int_a^b \mathrm{t}\mathcal{N}(x \,|\, \mu, \sigma^2, a, b) \log \frac{\mathrm{t}\mathcal{N}(x \,|\, \mu, \sigma^2, a, b)}{\mathcal{U}(x \,|\, a, b)} dx =$$

$$= -\mathcal{H}(\mathrm{t}\mathcal{N}(x \,|\, \mu, \sigma^2, a, b)) + \log(b - a)$$

Entropy for the truncated normal distribution is

$$\mathcal{H}(\mathrm{t}\mathcal{N}(x \,|\, \mu, \sigma^2, a, b)) = \log(\sqrt{2\pi e}\sigma Z) + \frac{\alpha\phi(\alpha) - \beta\phi(\beta)}{2Z}$$

$$\phi(x) = \frac{1}{\sqrt{2\pi}} \exp\left(-\frac{1}{2}x^2\right)$$

$$\Phi(x) = \frac{1}{2}(1 + \mathrm{erf}(x/\sqrt{2}))$$

Finally, we obtain

$$\mathrm{KL}(q(\theta_i \,|\, \mu_i, \sigma_i) \,\|\, p(\theta_i)) =$$

$$= \log(b - a) - \log(\sqrt{2\pi e}\sigma_i) - \log(\Phi(\beta_i) - \Phi(\alpha_i)) - \frac{\alpha_i\phi(\alpha_i) - \beta_i\phi(\beta_i)}{2(\Phi(\beta_i) - \Phi(\alpha_i))}$$

## B    Sampling from the truncated log-normal distribution

To sample $\theta^i$ from the truncated log-normal distribution, we first sample $\log \theta^i \sim \mathrm{t}\mathcal{N}(\log \theta^i \,|\, \mu_i, \sigma_i^2, a, b)$ and then take the exponent. In order to sample from the truncated normal distribution we use the inverse cumulative density function. The CDF for the truncated normal distribution is

$$F(x) = \frac{\Phi\left(\frac{x - \mu}{\sigma}\right) - \Phi\left(\frac{a - \mu}{\sigma}\right)}{Z} = y$$

Hence inverse CDF can be written as

$$F^{-1}(y) = \mu + \sigma\Phi^{-1}\left(\Phi(\alpha) + Zy\right) = x$$

Sampling $y$ from $\mathcal{U}[0, 1]$ one can obtain $x$ samples from truncated normal distribution. The final expression for the reparameterization trick for $\theta \sim \log \mathrm{t}\mathcal{N}(\theta \,|\, \mu, \sigma^2, a, b)$ looks like this:

$$\theta = \exp\left\{\mu + \sigma\Phi^{-1}\left(\Phi(\alpha) + Zy\right)\right\}, \text{ where } y \sim \mathcal{U}(y \,|\, a, b)$$

## C    Mean of the truncated log-normal ditribution

Let's derive the expected value of $\theta$. In order to this, let's first find the PDF of the truncated log-normal distribution.

$$\mathrm{t}\mathcal{N}(x \,|\, \mu, \sigma^2, a, b) = \frac{1}{Z\sqrt{2\pi\sigma^2}} \exp\left(-\frac{1}{2\sigma^2}(x - \mu)^2\right), \ x \in [a, b],$$

$$\mathrm{t}\mathcal{N}(\log x \,|\, \mu, \sigma^2, a, b)d(\log x) = \mathrm{t}\mathcal{N}(\log x \,|\, \mu, \sigma^2, a, b)\frac{dx}{x} = \mathrm{LogN}_{[a,b]}(x \,|\, \mu, \sigma^2)dx$$

$$\mathrm{LogN}_{[a,b]}(x \,|\, \mu, \sigma^2) = \frac{1}{Zx\sqrt{2\pi\sigma^2}} \exp\left(-\frac{1}{2\sigma^2}(\log x - \mu)^2\right), \ \log x \in [a, b], \ x > 0$$

The obtained PDF is very similar to the log-normal distribution PDF. Hence,

$$\mathbb{E}\Big[x \sim \mathrm{LogN}_{[a,b]}(x \mid \mu, \sigma^2)\Big] = \frac{1}{Z} \int_{e^a}^{e^b} \frac{1}{\sqrt{2\pi\sigma^2}} \exp\Big(-\frac{1}{2\sigma^2}(\log x - \mu)^2\Big)dx =$$

$$= \frac{1}{Z}\Bigg[\exp(\mu + \sigma^2/2) - \int_0^{e^a} \frac{1}{\sqrt{2\pi\sigma^2}} \exp\Big(-\frac{1}{2\sigma^2}(\log x - \mu)^2\Big)dx -$$

$$- \int_{e^b}^{+\infty} \frac{1}{\sqrt{2\pi\sigma^2}} \exp\Big(-\frac{1}{2\sigma^2}(\log x - \mu)^2\Big)dx\Bigg]$$

Now we need the formula for

$$p(a) := \int_a^{\infty} \frac{1}{\sqrt{2\pi\sigma^2}} \exp\Big(-\frac{1}{2\sigma^2}(\log x - \mu)^2\Big)dx$$

$$t = \frac{\log x - \mu}{\sigma}, \quad x = e^{t\sigma + \mu}, \quad dx = e^{t\sigma + \mu}\sigma dt$$

$$p(a) = \int_{\frac{\log a - \mu}{\sigma}}^{\infty} \frac{1}{\sqrt{2\pi}} \exp\Big(-\frac{1}{2}t^2 + t\sigma + \mu\Big)dt = \int_{\frac{\log a - \mu}{\sigma}}^{\infty} \frac{1}{\sqrt{2\pi}} \exp\Big(-\frac{1}{2}(t^2 - 2t\sigma + \sigma^2) + \frac{\sigma^2}{2} + \mu\Big)dt =$$

$$= \exp(\sigma^2/2 + \mu) \int_{\frac{\log a - \mu}{\sigma}}^{\infty} \frac{1}{\sqrt{2\pi}} \exp\Big(-\frac{1}{2}(t - \sigma)^2\Big)dt = \exp(\sigma^2/2 + \mu)\int_{\frac{\log a - \mu}{\sigma} - \sigma}^{\infty} \frac{1}{\sqrt{2\pi}} \exp\Big(-\frac{1}{2}t^2\Big)dt$$

$$p(a) = \exp(\sigma^2/2 + \mu)\Big(1 - \Phi\Big(\frac{\log a - \mu}{\sigma} - \sigma\Big)\Big) = \exp(\sigma^2/2 + \mu)\Phi\Big(\frac{\sigma^2 + \mu - \log a}{\sigma}\Big)$$

Using the derived formula, we finally obtain the expectation $\mathbb{E}x$

$$\mathbb{E}\Big[x \sim \mathrm{LogN}_{[a,b]}(x \mid \mu, \sigma^2)\Big] =$$

$$= \frac{1}{Z}\Bigg[\exp(\mu + \sigma^2/2) - [\exp(\mu + \sigma^2/2) - p(e^a)] - p(e^b)\Bigg]$$

$$= \frac{\exp(\mu + \sigma^2/2)}{Z}\Bigg[\Phi\Big(\frac{\sigma^2 + \mu - a}{\sigma}\Big) - \Phi\Big(\frac{\sigma^2 + \mu - b}{\sigma}\Big)\Bigg]$$

## D  Signal-to-noise ratio of the truncated log-normal distribution

It is useful to calculate the signal-to-noise ratio $\mathbb{E}x/\sqrt{\mathrm{Var}(x)}$ for the truncated log-normal distribution in order to investigate the sparsity of the resulting layer. We need the variance $\mathrm{Var}(x)$ to calculate it.

$$\mathrm{LogN}_{[a,b]}(x \mid \mu, \sigma^2) = \frac{1}{Zx\sqrt{2\pi\sigma^2}} \exp\Big(-\frac{1}{2\sigma^2}(\log x - \mu)^2\Big), \ \log x \in [a, b], \ x > 0$$

$$\mathrm{Var}\Big[x \sim \mathrm{LogN}_{[a,b]}(x \mid \mu, \sigma^2)\Big] = \frac{1}{Z}\int_{e^a}^{e^b} \frac{(x - \mathbb{E}x)^2}{x\sqrt{2\pi\sigma^2}} \exp\Big(-\frac{1}{2\sigma^2}(\log x - \mu)^2\Big)dx =$$

$$= \frac{1}{Z}\Bigg[\int_{e^a}^{\infty} \frac{(x - \mathbb{E}x)^2}{x\sqrt{2\pi\sigma^2}} \exp\Big(-\frac{1}{2\sigma^2}(\log x - \mu)^2\Big)dx - \int_{e^b}^{\infty} \frac{(x - \mathbb{E}x)^2}{x\sqrt{2\pi\sigma^2}} \exp\Big(-\frac{1}{2\sigma^2}(\log x - \mu)^2\Big)dx\Bigg]$$

So we have to calculate

$$p'(a) := \int_a^{\infty} \frac{(x - \mathbb{E}x)^2}{x\sqrt{2\pi\sigma^2}} \exp\Big(-\frac{1}{2\sigma^2}(\log x - \mu)^2\Big)dx$$

$$p'(a) = \int_a^{\infty} \frac{x}{\sqrt{2\pi\sigma^2}} \exp\Big(-\frac{1}{2\sigma^2}(\log x - \mu)^2\Big)dx$$

$$- 2\mathbb{E}x \int_a^{\infty} \frac{1}{\sqrt{2\pi\sigma^2}} \exp\Big(-\frac{1}{2\sigma^2}(\log x - \mu)^2\Big)dx$$

$$+ (\mathbb{E}x)^2 \int_a^{\infty} \frac{1}{x\sqrt{2\pi\sigma^2}} \exp\Big(-\frac{1}{2\sigma^2}(\log x - \mu)^2\Big)dx$$

We already have the expression for $p(a)$

$$p(a) = \int_a^\infty \frac{1}{\sqrt{2\pi\sigma^2}} \exp\left(-\frac{1}{2\sigma^2}(\log x - \mu)^2\right) dx = \exp(\sigma^2/2 + \mu)\Phi\left(\sigma - \frac{\log a - \mu}{\sigma}\right)$$

For the rest two summands we introduce a new variable $t$.

$$t = \frac{\log x - \mu}{\sigma}, \quad x = e^{t\sigma + \mu}, \quad dx = e^{t\sigma + \mu}\sigma dt$$

Next, we use $t$ in a variable substitution.

$$\int_a^\infty \frac{x}{\sqrt{2\pi\sigma^2}} \exp\left(-\frac{1}{2\sigma^2}(\log x - \mu)^2\right) dx = \int_{\frac{\log a - \mu}{\sigma}}^\infty \frac{1}{\sqrt{2\pi}} \exp\left(-\frac{1}{2}t^2 + 2t\sigma + 2\mu\right) dt =$$

$$= \int_{\frac{\log a - \mu}{\sigma}}^\infty \frac{1}{\sqrt{2\pi}} \exp\left(-\frac{1}{2}(t^2 - 4t\sigma + 4\sigma^2) + 2\mu + 2\sigma^2\right) dt =$$

$$= \exp(2\mu + 2\sigma^2) \int_{\frac{\log a - \mu}{\sigma}}^\infty \frac{1}{\sqrt{2\pi}} \exp\left(-\frac{1}{2}(t - 2\sigma)^2\right) dt = \exp(2\mu + 2\sigma^2) \int_{\frac{\log a - \mu}{\sigma} - 2\sigma}^\infty \frac{1}{\sqrt{2\pi}} \exp\left(-\frac{1}{2}t^2\right) dt =$$

$$= \exp(2\mu + 2\sigma^2)\Phi\left(2\sigma - \frac{\log a - \mu}{\sigma}\right)$$

$$\int_a^\infty \frac{1}{x\sqrt{2\pi\sigma^2}} \exp\left(-\frac{1}{2\sigma^2}(\log x - \mu)^2\right) dx = \int_{\frac{\log a - \mu}{\sigma}}^\infty \frac{1}{\sqrt{2\pi}} \exp\left(-\frac{1}{2}t^2\right) dt = \Phi\left(-\frac{\log a - \mu}{\sigma}\right)$$

Using the expression for $\mathbb{E}x$ we obtain

$$p'(a) = \exp(2\mu + 2\sigma^2)\Phi\left(2\sigma - \frac{\log a - \mu}{\sigma}\right) - 2\exp(\sigma^2/2 + \mu)\Phi\left(\sigma - \frac{\log a - \mu}{\sigma}\right)\mathbb{E}x + (\mathbb{E}x)^2\Phi\left(-\frac{\log a - \mu}{\sigma}\right)$$

$$\begin{aligned}
\mathrm{Var}\left[x \sim \mathrm{LogN}_{[a,b]}(x \mid \mu, \sigma^2)\right] &= \frac{1}{Z}\int_{e^a}^{e^b} \frac{(x - \mathbb{E}x)^2}{x\sqrt{2\pi\sigma^2}} \exp\left(-\frac{1}{2\sigma^2}(\log x - \mu)^2\right) dx = \\
&= \frac{\exp(2\mu + \sigma^2)}{Z}\left[\exp(\sigma^2)(\Phi(2\sigma - \alpha) - \Phi(2\sigma - \beta)) \right.\\
&\quad \left. -\frac{2}{Z}(\Phi(\sigma - \alpha) - \Phi(\sigma - \beta))^2 + \frac{1}{Z^2}(\Phi(\sigma - \alpha) - \Phi(\sigma - \beta))^2(\Phi(-\alpha) - \Phi(-\beta))\right] \\
&= \frac{\exp(2\mu + \sigma^2)}{Z}\left[\exp(\sigma^2)(\Phi(2\sigma - \alpha) - \Phi(2\sigma - \beta)) - \frac{1}{Z}(\Phi(\sigma - \alpha) - \Phi(\sigma - \beta))^2\right]
\end{aligned}$$

Finally, we obtain the signal-to-noise ratio

$$\mathrm{SNR}(x) = \frac{\mathbb{E}x}{\sqrt{\mathrm{Var}(x)}} = \frac{\left[\Phi(\sigma - \alpha) - \Phi(\sigma - \beta)\right]}{\sqrt{Z\exp(\sigma^2)(\Phi(2\sigma - \alpha) - \Phi(2\sigma - \beta)) - (\Phi(\sigma - \alpha) - \Phi(\sigma - \beta))^2}}$$

# E  Stable Computation of Statistics

Straightforward computation of the SNR and the mean of the truncated log-normal distribution can lead to indeterminate values like $0 \cdot \infty$ when the values of $\sigma$ are high. In order to make our calculations stable we use the scaled complementary error function $\mathrm{erfcx}(x) = \exp(x^2)\mathrm{erfc}(x)$. Given the equation

$$\Phi(a) - \Phi(b) = \frac{1}{2}\left[\mathrm{erf}\left(\frac{a}{\sqrt{2}}\right) - \mathrm{erf}\left(\frac{b}{\sqrt{2}}\right)\right] = \frac{1}{2}\left[\mathrm{erfcx}\left(\frac{b}{\sqrt{2}}\right)\exp\left(-\frac{b^2}{2}\right) - \mathrm{erfcx}\left(\frac{a}{\sqrt{2}}\right)\exp\left(-\frac{a^2}{2}\right)\right],$$

we can rewrite equations (19), (20) in the following form

$$\mathbb{E}\theta = \exp(\mu)\frac{1}{2Z}\left[\operatorname{erfcx}\left(\frac{\sigma-\beta}{\sqrt{2}}\right)\exp\left(b-\mu-\frac{\beta^2}{2}\right) - \operatorname{erfcx}\left(\frac{\sigma-\alpha}{\sqrt{2}}\right)\exp\left(a-\mu-\frac{\alpha^2}{2}\right)\right]$$

$$\operatorname{SNR}(\theta) = \frac{1}{\sqrt{D}}\left[\operatorname{erfcx}\left(\frac{\sigma-\beta}{\sqrt{2}}\right)\exp\left(b-\mu-\frac{\beta^2}{2}\right) - \operatorname{erfcx}\left(\frac{\sigma-\alpha}{\sqrt{2}}\right)\exp\left(a-\mu-\frac{\alpha^2}{2}\right)\right],$$

$$D = 2Z\left[\operatorname{erfcx}\left(\frac{2\sigma-\beta}{\sqrt{2}}\right)\exp\left(2(b-\mu)-\frac{\beta^2}{2}\right) - \operatorname{erfcx}\left(\frac{2\sigma-\alpha}{\sqrt{2}}\right)\exp\left(2(a-\mu)-\frac{\alpha^2}{2}\right)\right]$$
$$- \left[\operatorname{erfcx}\left(\frac{\sigma-\beta}{\sqrt{2}}\right)\exp\left(b-\mu-\frac{\beta^2}{2}\right) - \operatorname{erfcx}\left(\frac{\sigma-\alpha}{\sqrt{2}}\right)\exp\left(a-\mu-\frac{\alpha^2}{2}\right)\right]$$