[Reviews · NeurIPS 2017]

Reviewer 1



Summary: The paper explores group sparsity in Bayesian neural networks. The authors use truncated log-uniform distributed random variables to turn off uninformative neurons/filters. Experiments demonstrating sparsity without significant generalization loss are presented. Clarity and Quality: The paper is sufficiently clear and easy to follow. The proposed group sparsity inducing priors are a natural extension of existing work interpreting Gaussian dropout as variational inference in BNNs. Truncation is a technically reasonable approach for dealing with issues stemming from the use of improper priors necessitated by this interpretation. Overall the paper is technically sound. Originality and Significance - The primary contribution of the paper is the use of truncated log-uniform priors over scale (noise) variables to induce group structured sparsity in BNNs. Automatic model selection in BNNs is an interesting problem and being able to turn off uninformative neurons and filters is a useful step in that direction. Other Comments: 1) The choice of dealing with improper log-uniform prior seems cumbersome. Alternate, sparsity inducing proper priors such as continuous relaxations of the spike and slab (Horseshoe variants) are more convenient to work with. Under an appropriate parameterization, they replace the hard constraints on the range of theta [0, 1], with a smooth penalty and hence an easier optimization problem. These alternate priors do weaken the connection to the variational interpretation of dropout, but so does truncation and the connection to variational dropout is not necessary for structured sparsity. 2) While it may be a common computational trick, summarizing the posterior with a point estimate for test predictions is fraught with difficulties when the posterior variance hasn’t collapsed to (nearly) zero. It will be useful to include an analysis of the posterior variances of theta_i, to ascertain the quality of approximation of Eqn. 18. In any case, the median is perhaps a better summary of the skewed log-normal distribution than the mean and may be worth exploring in place of Equation 18. 3) The discussion on Page 4 justifying the use of log-Normal family of variational approximations for theta is bordering on unnecessary. While Gaussian variational approximations are commonly used for approximating the distributions of rvs with support over the entire real line no sensible variational inference scheme uses Gaussian approximations for rvs with support on only the positive half of the real line. Log-Normals are a popular and well-known choice for such random variables.

Reviewer 2



Summary: The paper addresses the actual problem of compression and efficient computation of deep NNs. It presents a new pruning technique for deep NNs, which is an enhancement of the Bayesian dropout method. Contrary to standard Bayesian dropout, the proposed method prunes NNs in a structured way. The NN-model is altered by dropout layers with an advanced probabilistic model (with truncated log-uniform prior distribution and truncated log-uniform posterior distribution). For the altered model, the authors propose a new "signal-to-ratio" pruning metric applicable to both individual neurons and more complex structures (e.g., filters). The submission and supplementary files contain a theoretical explanation of the method and derivation of the corresponding formulas. Moreover, an experimental evaluation of the approach is presented, with promising results. Quality: The paper has very good technical quality. It contains a thorough theoretical analysis of the proposed model and its properties. It also presents several experimental results, that evaluate the qualities of the method and compare it with two concurrent approaches. Clarity: The presentation is comprehensible and well-organized, the method is described in detail including derivation of formulas and implementation details. Originality and significance: The paper addresses the actual problem of compression and efficient computation of deep NNs. Its main contribution is the presentation of a new general technique for structured pruning of NNs that is based on Bayesian dropout. Contrary to the original method, 1) it uses more sophisticated and proper noise distributions and it involves a novel pruning metric, 2) the subject of pruning are not weights, but more complex structures (e.g., filters, neurons). 3) The method can be applied to several NN architectures (CNN, fully-connected,...), and it seems that it can be easily added to existing NN implementations. The presented approach proved in the experiments to massively prune the network and to speed up the computation on GPU and CPU, while inducing only small accuracy loss. However, based on the experimental results presented, the benefit over the concurrent "SSL" approach is relatively small. Typo: ...that is has ... (line 117) Comments: - Table 1 and Table 2 contain rows with results for concurrent methods called "SSL" and "SparseVD", without a short description of the main principles of these methods in the text. - It would be useful to assess also the efficiency of the pruning process itself. - The method has several parameters that need to be set. An interesting question is, how does the actual setting of these parameters affect the balance between sparsity and accuracy of the final model.

Reviewer 3



The authors proposes a new dropout variant which is demonstrated in a neural network unit-pruning application, carried out by learning the dropout rates under a sparsity-inducing prior. The methodology follows that of recent Bayesian approaches to deep learning, and the application is similar to that of [1] (which has some fundamental limitations stemming from the improper prior and loose KL approximations). The main novelty of the present paper is the use of a truncated log uniform prior instead of [1]'s log uniform prior, and truncated log normal approximating posterior (instead of [1]'s Gaussian approximating distribution). The authors justify why the choice of the prior and approximating distribution is sensible, and give an analytical KL which does not diverge to infinity unlike [1]'s KL. It is worth noting though that the authors do not place a prior distribution over the weights, but rather place a prior over latent "dropout" variables. Ie the model the authors perform inference in is not a Bayesian neural network (like in [1,2]) but rather a latent variable model (like [3]). Major comments to the authors: * Please clarify in the introduction what the prior is placed over (ie over auxiliary latent variables); it was not clear until page 3. * The literature survey is lacking many related works such as [2, 3, 4] ([3] being most directly related to the authors' latent variable model). * I would expect to see a comparison to [1] by removing units with large alpha parameter values as a baseline. Minor comments to the authors: * The Random Labels experiment is rather neat * Placing dropout after each convolution layer to enforce sparsity over kernel-patch pairs has been proposed before by [5] * Eq. (4) - it is not clear what "max_phi" is * Missing parenthesis on line 147 * Appendix line 8: I assume x = log theta? There also seems to be a mistake in the first line. References: [1] Diederik P Kingma, Tim Salimans, and Max Welling. Variational dropout and the local reparameterization trick. In NIPS. Curran Associates, Inc., 2015. [2] Yarin Gal and Zoubin Ghahramani. Dropout as a Bayesian approximation: Representing model uncertainty in deep learning. ICML, 2016. [3] Shin-ichi Maeda. A Bayesian encourages dropout. arXiv preprint arXiv:1412.7003, 2014. [4] S Wang and C Manning. Fast dropout training. ICML, 2013. [5] Yarin Gal and Zoubin Ghahramani. Bayesian convolutional neural networks with Bernoulli approximate variational inference. ICLR workshop track, 2016.